# Chrononutrition—When We Eat Is of the Essence in Tackling Obesity

**DOI:** 10.3390/nu14235080

**Published:** 2022-11-29

**Authors:** Maninder Kaur Ahluwalia

**Affiliations:** Cardiff School of Sport and Health Sciences, Cardiff Metropolitan University, Cardiff CF5 2YB, UK; mahluwalia@cardiffmet.ac.uk

**Keywords:** obesity, circadian clocks, time restricted feeding, energy homeostasis

## Abstract

Obesity is a chronic and relapsing public health problem with an extensive list of associated comorbidities. The worldwide prevalence of obesity has nearly tripled over the last five decades and continues to pose a serious threat to wider society and the wellbeing of future generations. The pathogenesis of obesity is complex but diet plays a key role in the onset and progression of the disease. The human diet has changed drastically across the globe, with an estimate that approximately 72% of the calories consumed today come from foods that were not part of our ancestral diets and are not compatible with our metabolism. Additionally, multiple nutrient-independent factors, e.g., cost, accessibility, behaviours, culture, education, work commitments, knowledge and societal set-up, influence our food choices and eating patterns. Much research has been focused on ‘what to eat’ or ‘how much to eat’ to reduce the obesity burden, but increasingly evidence indicates that ‘when to eat’ is fundamental to human metabolism. Aligning feeding patterns to the 24-h circadian clock that regulates a wide range of physiological and behavioural processes has multiple health-promoting effects with anti-obesity being a major part. This article explores the current understanding of the interactions between the body clocks, bioactive dietary components and the less appreciated role of meal timings in energy homeostasis and obesity.

## 1. Introduction

Obesity is defined as a BMI (kg/m^2^) of ≥30 (further classified into class I (BMI 30.0 to 34.9), class II (BMI 35 to 39.9) and class III (BMI > 40.0) and a BMI between 25.0–29.9 is classified as overweight [1,2]. The World Health Organisation defines obesity as a disease and not just the biggest risk factor for the development of multiple non-communicable chronic diseases, such as metabolic syndrome, diabetes, hypertension, cardiovascular diseases and cancer [3]. The cost to treat obesity and associated diseases puts an immense pressure not only on the healthcare sector but also wider society. The rate of obesity is rising globally inexorably and most projections indicate that without a concerted action, by 2035 over 45% of the global population will be either overweight or obese [4].

The pathogenesis of obesity is complex, with multifaceted interactions between an individuals’ genetics and the environment. Although evidence suggests that there is 40–70% heritability for obesity, environmental factors, particularly diet and physical activity, are critical to its onset and progression [5]. At its simplest, obesity is a result of positive energy balance and the relatively lower cost and abundance of energy-dense foods occurring concomitantly with an increasingly sedentary lifestyle have driven the obesity pandemic. In many modern societies, a 24/7 work and social lifestyle has become the norm, leading to erratic sleep and food-consumption patterns, disrupting the harmony between the biological day/active phase and metabolic processes and further fuelling the crisis [6,7]. 

There is no simple solution to address the escalating obesity epidemic. Lifestyle interventions have been the focus of weight-loss strategies with limited success and issues with long-term compliance. Body clocks, present in virtually each cell in the body, synchronise all physiological and biochemical processes to the external environment of light/dark cycles, temperature and food availability. The internal body clocks also coordinate the metabolic processes in a way that light/dark cycles are aligned with active/rest and anabolic/catabolic reaction phases and the loss of this harmony leads to adverse metabolic outcomes. The relatively novel concept of chrononutrition, the interactions between the food components and timing with circadian mechanisms, offers a promising target for designing sustainable weight-management strategies. 

## 2. Chronobiology 

Life evolved and adapted to the light/dark cycles that are a result of the earth’s rotation on its own axis. Organisms have an internal 24-hour circadian (from the Latin words circa diem, about a day) clock that adapts their daily activities to their external environment. Zeitgebers (external cues that synchronise the biological rhythms) such as light/dark cycles entrain (synchronise) the circadian system to generate rhythms in bodily processes, including the sleep/wake cycle, immune activity, metabolism, body temperature and blood pressure. 

At the molecular level, the circadian clock in mammals is composed of two sub-systems: the core and the peripheral clocks. The core circadian clock is situated in the anterior hypothalamus and includes the suprachiasmatic nucleus (SCN), composed of about 20,000 neurons, for which light is the primary zeitgeber. Light enters the eye and the photic signal is conveyed to the SCN, where it is integrated with non-photic signals that include food and external temperature [8]. An endogenous rhythm is generated and communicated to other parts of the brain and to peripheral organs via direct neuronal synaptic connections and endocrine signals, aligning the whole-body circadian clock to light. In addition to the core pacemaker, each cell in the body has its own local clock with its autonomous daily rhythmicity. These peripheral clock systems are influenced by the SCN but are also entrained by SCN-independent zeitgebers such as meal timing, locomotor activity and body temperature [9,10]. During the night, the SCN also regulates the synthesis and release of melatonin by the pineal gland. Melatonin is a sleep-inducing hormone with 24-hr rhythmicity; its production is inhibited by light, hence its low circulatory levels during the day [11]. The levels of melatonin start to rise approximately 2–3 h prior to the habitual nocturnal sleep time that also coincides with the onset of dim light conditions in the evening. This is defined as dim-light melatonin onset (DLMO), and is a reliable marker for the circadian entrainment [12,13]. Melatonin exerts its biological effects by binding to melatonin receptors, human melatonin receptor 1 A and 1 B (MTNR1A and MTNR1B). Sleep is an important modulator for multiple metabolic and endocrine pathways, linking melatonin levels, sleep duration and quality with obesity [14,15]. Melatonin also inhibits glucose-mediated insulin secretion and effects free radical scavenging, thus contributing directly to regulation of metabolic and immune function [16]. 

### Genomics and Epigenomics of Chronobiology

An intricate programme of transcription-translation-posttranslational feedback loops controls the complex patterns of circadian rhythmicity in physiology and behaviour (Figure 1). During the day, the key players responsible for the oscillations of circadian rhythms include the circadian locomotor output cycles kaput (CLOCK) and brain and muscle Arnt-like protein-1 (BMAL1). These heterodimerise and CLOCK-BMAL1 binds to the E-box regulatory element in the promoter regions of multiple circadian genes. Primarily, these activate the expression of PER (Period PER*1,* PER*2* and PER*3)* and CRY (Cryptochrome CRY*1* and CRY*2*) genes. Upon translation and a time lag (by the end of the day), these PER and CRY proteins accumulate in the cytoplasm and once they reach certain level, these heterodimerize to form a repressor complex and translocate to nucleus where they inhibit CLOCK-BMAL1-mediated transcription. Towards the end of the night, there is a gradual degradation of PER and CRY proteins that leads to the release of the CLOCK-BMAL1 dimer from PER/CRY suppression, and this re-initiates the clock cycle by inducing the transcription of PER and CRY. This cycle of activation and repression results in a feedback loop that generates an oscillation pattern of PER and CRY proteins over a 24-h period [9,17,18]. 

A parallel secondary loop exists that improves the robustness of the primary loop, consisting of the transcriptional factors nuclear receptor subfamily 1, group D, member 1 (NR1D1 gene producing a protein called REV-ERBα) and retinoic-acid-receptor-related orphan receptor α (RORα). The transcription of these genes is also activated by CLOCK-BMAL1 through binding to the E-box element in their promoters. REV-ERBα and RORα then compete to bind to the ROR element (RORE) in the BMAL1 promoter and regulate BMAL1 transcription; REV-ERBα suppresses while RORα stimulates BMAL1 transcription. Two other members of the nuclear receptor family, peroxisome proliferator-activated receptor α (PPARα), and PPARγ coactivator 1α (PGC1α) are activated by CLOCK-BMAL1. Interestingly, PPARα and PGC1α also activate BMAL1 transcription (Figure 1) [19,20,21,22]. 

The clock components are also under the control of various post-translational modifications including phosphorylation, ubiquitination, acetylation, poly-ADP-ribosylation and proteasomal degradation. This additional layer of regulation allows plasticity in the circadian system, making them highly responsive to an organism’s environment [23]. PER, CRY and PGC1α are all modified in response to the nutritional status of the cell, have an impact on the inhibitory period on CLOCK-BMAL1 and contribute to the changes in rhythmic patterns. CLOCK is an acetyltransferase and acetylates its partner BMAL1 and regulates the transcriptional activity of the heterodimer. CLOCK itself can be modified by poly-ADP-ribosylation leading to the transcriptional inhibition of CLOCK-BMAL1 target genes and altering the circadian cycle [24,25,26]. 

Circadian regulation is not limited to the core clock-network genes mentioned above; approximately 10–40% of the rodent genes in a tissue and >80% of the protein coding genes in baboons have been found to exhibit 24-hr rhythmic oscillations, resulting in identification of these genes as clock-controlled genes, “CCGs” [27]. These CCGs regulate various biological processes including apoptosis, metabolism, detoxification, cell-cycle regulation and immune function [28,29,30]. Some genes are under the control of SCN-generated rhythms, but a large proportion of genes are influenced by tissue-specific peripheral clocks. It has been shown that in mouse liver, 90% of the transcripts showing circadian oscillations are under the influence of local clock machinery [31]. 

The oscillations produced are characterised by their amplitude (magnitude of cycle reflecting the strength of the rhythm), period (the time interval between two recurring waves within a rhythm), phase (any time point on a rhythmic cycle, e.g., peak relative to an external reference such as midnight) and MESOR (midline estimating statistic of rhythm). These features indicate the rhythmicity and robustness of the circadian clock and any changes in these parameters can be a predictor of health-related outcomes (Figure 1) [32,33,34]. Secondly, the rhythms generated by the core and peripheral clock systems need to be aligned given that misalignment between the clocks can potentially disrupt the body’s physiological patterns and metabolism. 

## 3. Chronotypes, Chronodisruption and Energy Homeostasis

The circadian cycle for humans has an average period of 24.2 h, but this period varies considerably between individuals and is defined as their chronotype [35]. Chronotypes range between early birds (morning people/advanced sleep phase (ASP)/early chronotypes, people going to bed and waking up early, circadian period shorter than 24.2 h) to night owls (evening people/delayed sleep phase (DSP)/late chronotypes, people preferring late bedtime and waking up late, a circadian period longer than 24.2 h). It is estimated that about 40% of the population can be classified into either of these extremes [35,36]. There is a continuum between these two extreme phenotypes, and individuals within this category are referred to as belonging to the intermediate or neutral chronotype; about 60% of the population falls into this category [37]. Melatonin rhythms and DLMO vary by as much as 2 h between the chronotypes [38]. Certain rare and genetic forms of extreme chronotypes have also been recognised. A type of insomnia where sleep patterns are shifted to a delayed onset and offset times compared to the societal norm is known as delayed sleep phase disorder (DSPD), while a phenotype associated with habitual sleep times that are earlier than the solar morning or societal norm is termed familial advanced sleep phase disorder (FASPD) [39]. In addition to the age, gender and societal set-up, an individual’s chronotype is also influenced by their genetic makeup and various genome-wide association studies (GWAS) and candidate-gene approaches have associated more than 350 loci with the morning chronotype and, not surprisingly, these include the components from the clock machinery [40]. Genetic variants within the clock machinery have been associated with sleep patterns, variation in energy intake, waist circumference, obesity and metabolic diseases (Table 1) [41,42,43]. Both morning and evening chronotypes are multigenic and are influenced by the environment, whereas non-genetic factors such as artificial light and social pressures contribute more to the evening chronotype [44]. Although these chronotypes result in a preferred choice of sleep and activity patterns, these do not directly contribute to the pathogenesis of metabolic diseases. However, some recent studies have suggested a link with these morningness or eveningness tendencies and metabolic health. The evening chronotype has been associated with unhealthy food choices, binge eating, night snacks and multiple metabolic disorders, including obesity, while morning individuals are associated with lower rates of depression and improved mental health [40,44,45,46,47,48,49,50]. The chronotype of an individual determines their sleep, dietary and activity patterns and although they indirectly influence the sleep duration and quality, these are distinct from sleep duration [51]. Any forced disruption to the normal sleep patterns, e.g., shift work or frequent traveling over two or more time zones (jetlag), can lead to circadian misalignment and have been associated with various metabolic diseases. An individual with the evening chronotype tends to go to bed late but due to external demands (occupational commitments such as working hours or school start time in case of children) would result in waking up to an alarm clock that is out of phase from their biological circadian cycle, resulting in shorter sleep duration. There can also be variation in bedtime within the week, e.g., weekdays vs. the weekends, leading to the concept of social jet-lag. These societal activities indirectly contribute to metabolic health (Table 1) [16,44]. 

There is an intricate and bidirectional relationship between the circadian clock and metabolism that contributes to the overall metabolic homeostasis. There needs to be an optimal alignment between central and peripheral clock requiring energy intake to be aligned with the active phase/biological day for diurnal organisms such as humans (and night-time for nocturnal animals such as rodents). Mice consume about 80% and humans approximately 100% of the nutrients during the wake/active phase [75]. This pattern is in-tune with the oscillations in metabolic pathways of primitive hunter-gatherer humans, who were exposed to feast/famine cycles that coincided with active/rest phases. During the active phase, when humans could forage and hunt for food, the energy intake was higher and the metabolic pathways were geared towards replenishing the energy stores. While famine, generally associated with rest meant that the body had to adapt to starvation or restriction in food intake and the metabolism would switch to catabolic processes and mobilisation of energy stores. Humans are genetically programmed to these oscillations in energy stores, which is incompatible with a modern lifestyle with the constant availability of high-energy foods.

These results, when combined with a sedentary lifestyle, blunt the oscillations and lead to metabolic disturbances with a plethora of associated diseases [76,77,78]. 

Misalignment in the active/rest and feast/famine phases can be due to endogenous factors, e.g., genetic variants in the core clock machinery or due to external lifestyle factors such as extended exposure to artificial light, increased shift work, sedentarism, untimely and frequent snacking and jetlag, and leads to chronobiological vulnerabilities to various diseases (Table 2) [46,79,80,81,82]. An umbrella term used for circadian disruptions is ‘chronodisruption’. The term has evolved since it was first coined in 2003, with short-term disturbance being called circadian disruption. In contrast, long-term disturbances leading to adaptations without a negative impact on health are termed chronodisturbance, and long-term desynchronisation contributing to disease is called chronodisruption [83]. Artificial light exposure, even at low levels such as from electronic devices including phones, also interferes with the DLMO and melatonin levels, sleep onset and duration [16,44,84]. Additionally, night eating, irregular eating patterns or feeding over the resting periods even in the absence of evening chronotype lead to misalignment and impact the robustness of the oscillations, compromising the metabolic homeostasis and leading to higher BMI and disease development (Figure 2) [76,85,86,87,88,89,90,91,92,93,94,95]. 

## 4. Chrononutrition

Understanding the molecular basis of chronodisruption can potentially allow us to develop practical strategies to improve circadian alignment to mitigate the burden of metabolic diseases. One such relatively novel approach is termed ‘chrononutrition’, encompassing two elements: dietary components that regulate circadian system and meal timings to synchronise misaligned molecular clocks, which can act either positively or negatively on metabolic activity. These interventions (including physical activity) can improve the blunted rhythmic oscillations; even if these are not as robust as those accompanying feast/famine cycles, they can potentially be associated with positive metabolic health outcomes (Figure 2) [111].

### 4.1. What to Eat

It is well-established that nutritional components, including macronutrients and natural bioactive compounds, have the ability to (directly or indirectly) regulate the expression of various genes, and clock-network genes are no exception [112,113,114,115]. While feeding/fasting patterns mainly act as a potent zeitgeber for peripheral clocks and minimally impact the master clock, nutritional components are also able to influence the master clock in the SCN [116,117,118]. High-fat diets are the best-known circadian rhythm disruptors and can lead to the reversal of feeding patterns and perturbed metabolic parameters [117,119,120,121,122]. The relative distribution of macronutrients in diet can also contribute to central and peripheral clock modulation in humans [115,123]. Other nutrients have been investigated for their role in circadian remodelling; a ketogenic diet increases the activation of CCGs via CLOCK-BMAL1 activation, high sodium and high salt intake causes a phase delay in BMAL1 and CRY1 and PER2 peak expression and caffeine and theophylline lengthen the period of the cellular circadian clock [124,125,126,127]. 

A growing body of evidence is emerging that links the use of natural bioactive compounds to health via synchronising or improving circadian rhythmic patterns and potentially acting as a natural chronobiotic—an agent with the ability to adjust the timing of one’s internal biological rhythm. The best studied chronobiotic is melatonin, which when administered exogenously can shift the circadian clock phase and alter circadian rhythms in endogenous hormones, body temperature and behaviour [128]. There is also evidence that melatonin supplementation not only modulates body weight and metabolic parameters but also has the ability to reverse the metabolic perturbations caused by chronodisruption [129,130]. Most of the melatonin supplementation has been in a synthetic form through capsules, but melatonin also exists in natural food sources such as fish, eggs, poultry, milk, nuts, fruits and seeds [131]. Natural plant derivatives such as phytochemicals, plant bioactives and nutraceuticals have gained significant attention for their health-promoting properties. Plant polyphenols are one of the most abundant and widely distributed group of secondary metabolites driven from plants. A diverse range of polyphenolic compounds, including phenolic acids, flavanones, flavonoids, tannins, lignans, stilbenes and curcuminoids, have been associated with anti-obesity activities via a variety of mechanisms [132,133,134,135]. Although the exact mechanisms of their actions remain unclear and there are issues around their absorption, bioavailability and bio-accessibility, evidence suggests that some of the beneficial effects of these compounds are mediated by their ability to interact with circadian clocks via genetic/epigenetic mechanisms or indirectly via altering the gut microbiota (Table 3) [136,137]. These interactions are complex; phytochemical content from the plant source depends on various agricultural factors such as soil, light, season, temperature and even the endogenous circadian clock of the plant [138]. There are also seasonal factors, e.g., availability, polyphenolic composition from the same source and human-consumption patterns, which can add another layer of complexity to the seasonal biological oscillations over the period of 12 months, called circannual rhythms [138,139,140]. The timing of consumption of these compounds is also critical, as demonstrated by a study using tryptophan-enriched milk formula in infants. Infants taking tryptophan-enriched formula during the night had improved sleep parameters as compared to those who consumed it during the day [141]. Table 3 presents some of the direct interactions observed between the polyphenolic compounds, circadian mechanisms and health outcomes. 

### 4.2. When to Eat

Interestingly, other than what goes on your plate, chrononutrition also highlights the significance of aligning the meal timing, frequency and the patterns of energy intake with the circadian rhythm [167]. The concept of ‘when you eat’ was first proposed in 1967 as a link between the meal timing, energy metabolism and chronic diseases by Franz Halberg [168,169]. Food consumption is a strong entrainer for peripheral circadian clocks. Optimal health requires an alignment of energy intake with the biological day and active phase and to generate feed/fast cycle that human physiology is adapted to. The transition between the feed/fast cycles requires a different set of transcription factors and associated proteins, which display diurnal variation. Genes that are active and peak during the day are mainly associated with glycogenesis and lipogenesis, with an overall aim of replenishing the energy stores, while the fasting phase is enriched with genes responsible for growth, repair, glycogenolysis and lipolysis. Any perturbations in the availability of the key players in either of these phases and dietary intake could lead to the dysregulation of energy metabolism [21,170,171]. The same meal consumed at different times during the circadian cycle could have a varied impact on energy metabolism. The current 24/7 lifestyle and a constant supply of nutrients interrupts human circadian physiology. Emerging data suggest that the eating window for more than 50% of the population (non-shift-work) is approximately 15 h a day, with less than 30% of energy consumption happening before noon and 30–45% of daily energy consumed during dinner and post-dinner snacks and part of it spanning over the circadian rest period [172,173,174]. The increased eating window and shorter overnight fast contributes to increased energy intake. Mistimed eating accompanied by erratic sleep patterns leads to dampened circadian rhythms and increases the risk of metabolic disorders (Table 2). Interestingly, the dampening of circadian rhythms by a high-fat diet can be recovered from by just limiting food intake during the biological active phase, highlighting the importance of “when to eat” and aligning meals with our biological clocks [175]. 

Time-restricted feeding (TRF) in animals (time-restricted eating (TRE) in humans) is an approach that aims to align meal times with the circadian rhythm and has gained significant attention in recent years. Multiple animal and human studies have been conducted and some human feasibility studies and clinical trials are summarised in Table 4. 

## 5. Time-Restricted Eating—Just Another Approach to Reduce Caloric Intake or a Circadian Alignment Tool

The role of calorie consumption in energy homeostasis is not disputed, and creating a negative energy balance is a logical approach to tilt the scales. Calorie restriction (CR) refers to a consistent dietary regimen low in calories, generally a daily 20–40% reduction as compared with ad libitum feeding, without malnutrition (Figure 3). This is not simply another term for fasting, which is commonly defined as the total abstinence from energy-containing foods and beverages for periods ranging from 12 h to 3 weeks, although some protocols employ modified fasting in which a minimal number of calories may be consumed [207]. CR is one of the most effective interventions for weight loss, improving health parameters in animals including primates and is a highly successful strategy for reducing age-related diseases and extending the mean and maximum lifespan in multiple species [208,209,210]. In addition to animal studies including mice and monkeys, CR over a 6-year period in a cohort of 18 participants showed improved BMI, glucose homeostasis and lipid profile and reduced inflammatory markers and blood pressure [211,212,213]. Although short-term caloric restriction is associated with 5–10% weight loss, long-term compliance is a massive challenge and there is a tendency to regain the lost weight [214,215]. To overcome these challenges, alternative dietary strategies such as intermittent caloric restriction have gained attention. 

The intermittent fasting (IF) approach involves introducing intermittent periods of eating deprivation, providing a less restrictive alternative to CR. The regimen includes periods of fasting where the energy restriction ranges from 60–100%, interrupted by periods of normal dietary intake (isocaloric/ad libitum). This approach is adopted in a variety of protocols; alternate day fasting (no calories on fasting day and ad libitum on feast days), alternate-day modified fasting (consuming <25% of usual caloric intake on fasting days and ad libitum on feast days), 5:2 diet (with 2 days of fasting with 60–100% energy restrictions and 5 days of isocaloric intake), 4:3 (with 3 days of fasting with 60–100% energy restrictions and 4 days of isocaloric intake) (Figure 3) [214,216,217,218]. Certain religious fasting practices observed including the Islamic month of Ramadan have been studied as part of IF approaches [219]. This approach, as compared to continuous CR, introduces periods of fasting when the metabolism shifts towards the catabolic state and mimics the feast/fast physiology of our hunter-gatherer past [220]. Even though the IF regimen suggests ad libitum feeding on non-fasting days, there is no full compensation for the fasting days/time, and overall there is an energy deficit or lack of calories. Various studies have compared IF approaches to continuous CR and reported comparable or better weight loss and improvement in metabolic health [189,221,222,223,224,225,226,227,228]. 

Time-restricted feeding (TRF) is considered a modified version of IF, where the energy intake is limited to a window of 4 to 12 h in order to extend the time spent in the fasted state regularly, without changes in caloric intake (Figure 3) [229]. Even though caloric restriction is not intentional in TRF, multiple studies have reported that restricting the feeding window to 8 h produces a mild caloric deficit [176,179]. In fact, any restriction to the eating window helps reduce the energy intake, e.g., just stopping night-time eating in healthy individuals leads to a reduction in energy intake [187]. Considering the deleterious effects of chronodisruption, extended eating duration and the imbalanced spread of energy intake during the day, it is clear that the timing of a meal is instrumental in fine tuning the energy balance. Thus, TRF is more than just a mode for caloric restriction or IF; it also synchronises the feeding time with the awake/active phase when the body is best able to metabolise it. More recently, the health-promoting role of caloric restriction has been shown to be partly due to TRF rather than just caloric intake and extended periods of fasting independent of caloric content share the same if not better health outcomes [230]. This leads to an alignment of the feeding-fasting cycle with circadian rhythms and offers a promising dietary strategy to mitigate the deleterious effect of chronodisruption [231]. Mice fed with a high-fat diet showed a dampened diurnal rhythm in physiology, which was recovered in the cohort on same diet but over a time-restricted period [232]. There is a plethora of research supporting TRF being beneficial in not only reducing body weight but also improving metabolic health in general (Table 4) [184,204,231,232,233,234]. Interestingly, as well as the length of the restriction window, the timing of the TRF within the 24-hour cycle is important and may provide slightly different outcomes. Restricting the feeding period to earlier in the day (eTRF) provides advantageous outcomes than mid-day TRF (mTRF) or later TRF (lTRF), as this aligns better with circadian biology, though larger studies and more data are required to fine-tune the interventions [204]. Attempts have also been made to have a pragmatic approach to adapt TRF approaches to life/work schedules. It has been shown that TRF for 8–9 h a day for 5 days and ad libitum for 2 days, still reverses or restricts diet-induced obesity [235,236]. 

Caloric restriction and intermittent fasting are not strictly part of the chrononutrition strategy, as the focus is not about aligning meal times with the biological clock. However, due to the common mechanisms of an overall negative energy balance, which is involuntary in IF and TRF, they share certain molecular mechanisms that contribute to the overall energy homeostasis. Each of the above-mentioned strategies have challenges and potential barriers to adherence for a long-term weight-loss strategy, though the current view supports TRF as a promising tool with greater-than-ordinary adherence, a good safety profile, and socially acceptable flexible implementation [237,238,239,240]

## 6. Mechanisms of Chrononutrition in Energy Homeostasis and Obesity

### 6.1. Appetite Control

Appetite, eating behaviour, hunger/satiety and energy homeostasis are controlled by the melanocortin system. This includes melanocortin receptor 4 (MC4R), which is present in the brain and is activated by its ligand, melanocyte-stimulating hormone (MSH). MSH is produced by the arcuate nucleus (ARC) in the hypothalamus that consists of two distinct types of neurons: anorexigenic neurons expressing proopiomelanocortin (POMC) the orexigenic neurons expressing NeuroPeptide-Y (NPY) and agouti-related protein (AgRP), having opposite effects on energy homeostasis. Leptin, an adipocyte-derived satiety hormone, activates POMC neurons, and its circulatory levels directly relate to adiposity. This results in the proteolytic conversion of POMC and the release of α-MSH, which activates MC4R, promoting satiety, reduced food intake and increased energy expenditure. Leptin also binds to AgRP neurons, resulting in the suppression of AgRP release, which is a potent antagonist for MC4R and increases food intake, energy conservation and weight [241,242]. An incretin hormone GLP1 (glucagon-like peptide-1) that stimulates insulin secretion, and PYY (peptide YY), both secreted from the gastrointestinal tract, are also anorexigenic and delay gastric emptying and promote satiety [243]. As opposed to leptin, ghrelin is an orexigenic hormone mainly derived from the stomach, which promotes hunger via activating AgRP neurons, which increases appetite and decreases energy expenditure (Figure 4) [244].

Leptin and ghrelin levels both exhibit diurnal oscillations and are influenced by obesity and food intake [245,246,247,248,249]. Circulating leptin levels peak at night and are lowest in the afternoon, but in obese subjects the amplitude in these oscillations is lost [249]. Circadian disruption abolishes circadian oscillation patterns of plasma leptin and induces leptin resistance [104,247,250]. Leptin is high in obese subjects, but, due to leptin resistance, the satiety signal is absent/compromised. Diets rich in fat inhibit the anorectic effects of leptin while sucrose- and fructose-rich diets promote leptin resistance [251,252]. Fasting leads to a drop in leptin levels and intermittent fasting improves leptin resistance. [253,254,255,256]. Leptin also regulates energy homeostasis through AMP-activated protein kinase (AMPK) by increasing fatty acid oxidation and reducing fatty acid biosynthesis [257,258]. Leptin also increases the expression of uncoupling protein-1(UCP1) and the browning of white adipose tissue and thermogenesis, discussed later in this section [259]. GLP1 and PYY are under the control of clock machinery and exhibit circadian patterns of rhythmicity [260,261]. 

Ghrelin levels increase during fasting or just before the habitual feeding time, dropping postprandially. Overweight and obese subjects lose this variation with a lesser drop in postprandial ghrelin levels leading to lower level of satiety after a meal, promoting snacking and overconsumption of food [262]. Ghrelin levels increase in response to caloric restriction and remain high for a considerable length of time, leading to increased food intake and regaining weight [263,264,265]. The response of plasma ghrelin levels to time-restricted feeding are inconsistent, with some studies reporting its reduction, while others show no effect [182,185,266,267]. 

Sleep duration also regulates ghrelin and leptin levels in circulation and contribute to energy homeostasis [268,269,270]. Insufficient sleep possibly works via altering the levels of melatonin, which plays a key role in food intake and energy expenditure. Melatonin reduces the expression of AgRP/NPY and increases the expression of POMC, hence regulating energy homeostasis via the MC4R pathway in the hypothalamus. Melatonin also inhibits leptin secretion and ameliorates leptin resistance, which accompanies obesity [249]. Lack of sleep also interferes with the weight loss achieved by caloric restriction, indicating the key role of sleep in terms of the efficacy of weight-loss strategies [269]. TRF and multiple natural bioactive compounds have been associated with an improvement in sleep [76,271,272]. 

### 6.2. Energy Sensors in the Body

#### 6.2.1. AMP-Activated Protein Kinase (AMPK)

AMP-activated protein kinase (AMPK) is the key energy sensor in the cell and has the ability to regulate whole-body metabolism. AMPK is activated upon a fall in intracellular ATP levels and increases in ADP or AMP levels, which reflects the energy status of the cell. 

Upon activation, AMPK switches on the catabolic pathways, leading to ATP generation and switching off the anabolic ATP-consuming pathways. ATP generation happens by promoting glycolysis and fatty acid oxidation and in the long term, by increasing mitochondrial content and the use of mitochondrial substrates as an energy source [273]. 

Fasting/intermittent fasting/nutritional deprivation activates AMPK, converting this nutritional signal to a circadian signal by phosphorylating CRY, resulting in its degradation. AMPK also phosphorylates casein kinase I epsilon, which in turn phosphorylates and degrades PER [274]. This removes repression on CLOCK-BMAL1, shortens the timing of the feedback loop and activates the transcription of the target genes including REV-ERBα, PER and CRY [275]. When a high-fat diet is administered ad libitum, this leads to a disturbed and dampened circadian rhythm of AMPK, while eTRF increases the amplitude of expression of AMPK [232,276]. 

One key target for activated AMPK is nicotinamide phosphoribosyltransferase (NAMPT), an enzyme that promotes an increase in intracellular levels of nicotinamide adenine dinucleotide (NAD^+^) levels (Figure 4). NAD^+^ is essential for cellular energy maintenance and central to cell health. NAD^+^ acts as a redox carrier that gets converted to NADH in various metabolic pathways including glycolysis, the TCA cycle and fatty acid oxidation. NADH serves as the hydride donor to the electron-transport chain for the production of ATP in mitochondria. Additionally, NAD^+^ acts as a cofactor or co-substrate to enzymes such as sirtuins and poly (ADP-ribose) polymerases (PARPs). All these processes continuously deplete the NAD^+^ pool in the cell, which can be replenished by de novo pathway from tryptophan and the predominant salvage pathway from the NAD^+^ degradation product nicotinamide (NAM) [277]. In the salvage pathway, NAM is converted by NAMPT, the rate-limiting enzyme in the pathway, to an intermediate product, nicotinamide mononucleotide (NMN). NMN adenyltransferase 1-3 (NMNAT1-3) then converts NMN into NAD^+^ [278]. The CLOCK-BMAL1 complex binds to the E-boxes in the NAMPT promoter and controls its transcription and the levels display robust circadian oscillations providing a 24-h rhythm to NAD^+^ levels in the cell. NAD^+^ levels peak approximately 4 h after the peak in NAMPT [279,280]. NAD^+^ levels depend on cellular energy levels. Glucose deprivation, fasting, caloric restriction and exercise lead to an increase, and high-fat diets decrease NAD^+^ levels [281,282,283]. Fasting or calorie restriction increases cellular NAD^+^ levels by activating NAMPT and feeding suppresses NAMPT, providing a link between metabolism and the circadian clock [21,280,283,284]. Sirtuins and PARPs (poly (ADP-ribose) polymerases) depend on and compete for the cellular pool of NAD^+^. Cellular NAD^+^ stores are important for cell health, delay the onset of multiple diseases and enhance longevity, and lifestyle interventions that lead to increased NAD^+^ bioavailability are recommended for positive health outcomes [285]. 

Sirtuins (silencing information regulator) are a seven-member superfamily of proteins, which deacetylases histones and non-histone proteins and use NAD^+^ as a co-substrate, converting it into nicotinamide (NAM). The dependence of sirtuins on NAD^+^ suggests their role as energy sensors of the cell [286]. Studies over the past two decades have provided strong evidence that sirtuins are the key mediators of the beneficial effects of caloric restriction [287,288,289,290,291]. NAD^+^ levels are increased in fasting, intermittent fasting or caloric restriction, leading to the activation of sirtuins. Activated sirtuins deacetylate a number of proteins playing key roles in metabolism, inflammation, autophagy, aging, apoptosis, oxidative stress, neurodegeneration and cancer [286,292]. SIRT1, the most extensively studied sirtuin, interacts with the clock machinery contributing to the circadian rhythms. SIRT1 expression follows circadian patterns mainly due to circadian regulation of NAMPT and NAD^+^ levels. In turn, SIRT1 binds to the CLOCK-BMAL1 complex in a circadian manner and regulates clock-dependent gene expression and oscillations, NAMPT being one of them (Figure 1) [293,294]. CLOCK is an acetyltransferase and acetylates BMAL1, which results in the activation of the dimer while SIRT1 counterbalances CLOCK activity and deacetylates BMAL1. Additionally, SIRT1 also deacetylates PER2 and enhances its degradation. The absence of SIRT1 leads to PER2 stabilisation and the inhibition of CLOCK-BMAL1 activity, impacting the expression of various clock-dependent genes while PER2 negatively regulates SIRT1 [293,294,295,296]. SIRT1, playing a dual role as an energy sensor and a regulator of clock components, couples metabolism and circadian mechanisms. SIRT1 activation also leads to deacetylation and enhanced PGC1α transcriptional activation [273,297,298,299]. 

In addition to caloric restriction, multiple dietary polyphenols have been investigated for their health-promoting properties via activating sirtuins—resveratrol being the most extensively researched (Table 3) [300,301,302]. In fact, a new term has emerged for food components that modulate sirtuin activities mainly by increasing the bioavailability of NAD^+^: ‘Sirtfood” [303,304,305,306]. 

Although a vast amount of data is available connecting sirtuins and caloric restriction, fasting and intermittent fasting, the data on the link between the time-restricted feeding and sirtuins have only started to emerge. In mice, TRF has been shown to increase SIRT1 expression and reverse the loss of circadian rhythm in SIRT1 expression caused by a high-fat ad libitum diet [200,307]. Human studies indicate that restricting eating time to the earlier part of the day upregulates SIRT1 expression and the amplitude of SIRT1 circadian oscillations and associated health benefits [184,200].

PARPs (poly (ADP-ribose) polymerases) catalyse the transfer of poly (ADP-ribose) from NAD^+^ to acceptor proteins to modulate their activity and deplete the cellular NAD^+^ stores. PARP1 is the best characterised member and responsible for about 90% of the total cellular poly-ADP-ribosylation activity. PARP1 plays a critical active role in DNA repair, metabolism, inflammation and cell death [285,308,309]. PARP1 expression exhibits circadian patterns and poly-ADP-ribosylates and modulates the activities of CLOCK-protein-inhibiting CLOCK-BMAL1 binding activity in a circadian manner [24]. Fasting reduces while a high-fat diet increases PARP1 protein and its activity [310]. The knockdown or inhibition of PARP1 in mice leads to leaner phenotype, increased availability of NAD^+^, activation of SIRT1 and related metabolic effects, while the reverse was observed in the case of a PARP1 inducer [310]. Loss of PARP1 contributes to the browning of white adipose tissue, partly by activating SIRT1 and PPARγ [311,312,313]. 

#### 6.2.2. Mitochondrial Dynamics

Mitochondria play a pivotal role in cellular metabolism by generating the basic unit of energy, ATP. Metabolic diseases such as obesity, a high-fat diet or excessive caloric intake all lead to mitochondrial dysfunction, which leads to energy deficiency and an increase in the production of reactive oxygen species, causing cell damage [314,315]. Cellular quality control processes including mitophagy and mitochondrial biogenesis ensure that mitochondrial capacity, function and integrity are preserved. Mitochondrial biogenesis is regulated by many transcriptional regulators present in the cell, PGC1α being the key one. PGC1α is an inducible transcriptional coactivator that regulates multiple transcription factors involved in energy metabolism and exhibits strong circadian rhythms in multiple tissues [316,317,318,319]. The activity of PGC1α is regulated by posttranslational modifications; the deacetylation and phosphorylation of PGC1α lead to its activation (Figure 4) [320,321].

Caloric restriction via fasting, intermittent fasting or time-restricted feeding leads to SIRT1 and AMPK activation, which leads to the deacetylation and phosphorylation of PGC1α, respectively; both these modifications activate PGC1α and an increased mitochondrial biogenesis [298,299,321,322,323]. The process of mitochondrial biogenesis is generally associated with improved cell health and has been reported in various cell types [324,325,326]. Inactive phase feeding in rats leads to an altered regulation of mitochondrial biogenesis [327]. Dietary interventions such as whole-grain bioactive compounds and various polyphenolic compounds also contribute to mitochondrial biogenesis, mainly via the SIRT- PGC1α pathway [328,329,330]. One of the tissues where mitochondrial biogenesis contributes to obesity and energy homeostasis is adipocytes. 

#### 6.2.3. Adipose Tissue 

There are two distinct categories of adipose tissue, white adipose tissue (WAT) and brown adipose tissue (BAT). WAT stores energy in the form of triglycerides in times of caloric excess and is associated with metabolic disease states. BAT is rich in mitochondria and has a high expression of uncoupling protein-1 (UCP1), can uncouple fatty acid oxidation from ATP production and is specialized for energy expenditure via thermogenesis, modulating the energy homeostasis. It has a protective role in metabolic health. BAT can also protect against diet-induced obesity, and genetic variants in key genes involved in the process have been recognised to contribute to impaired thermogenesis [331,332]. In addition to classical WAT and BAT, trans-differentiation of WAT can lead to a beige or brite adipose tissue via a process called ‘browning’, with positive health outcomes. These beige cells start to express UCP1 accompanied by mitochondrial biogenesis leading to an increased mass of mitochondria (Figure 4). PGC1α, PPARγ and PRDM16 are the three key modulators of browning of WAT and all have been associated with circadian clocks [333,334,335]. PPARγ is a nuclear transcription factor and a nutrient sensor that plays critical roles in adipocyte differentiation. PPARγ exhibits circadian rhythmicity at mRNA and protein level. Nocturin, a circadian-regulated gene also positively regulates PPARγ activity. PPARγ induces REV-ERBα expression and PER2 directly inhibits the expression of PPARγ [336]. PPARγ activation by synthetic ligands can upregulate browning in WAT and it requires the recruitment of PR (PRD1-BF1-RIZ1 homologous)-domain-containing 15 (PRDM16) to form the transcription complex. PRDM16 also induces the expression of PGC1α and UCP1 [337,338,339]. Additionally, the circadian components BMAL1, RORα and REV-ERBα also regulate BAT differentiation [335,340,341,342]. Thus, the modulation of the adipocyte phenotype to BAT contributes to enhanced energy homeostasis, and any strategy mediating this effect can positively contribute to health. 

Caloric restriction and exercise contribute to PGC1α activation and associated health outcomes [343,344,345,346,347,348]. Aligning meal times with the circadian rhythms by time-restricted feeding in animal models can also modulate PGC1α activity and mitochondrial biogenesis. [323,349,350,351,352,353]. Additionally, various polyphenolic compounds contribute to mitochondrial biogenesis via PGC1α activation, which is mediated via AMPK-SIRT1 activation [354,355]. Sleep, melatonin and leptin pathways also contribute to the browning of WAT and energy homeostasis via the induction of UCP1 expression and mitochondrial biogenesis (Figure 4) [249,259]. 

#### 6.2.4. Gut Microbiota

One of the mechanisms for health benefits from chrononutrition is via modulation of the gut microbial community. The gut is not just the site of nutrient absorption, it is also the home to a vast, complex, diverse microbial community. The delicate balance in gut flora diversity and its species/phyla distribution is critical for host health. In addition to dietary components, circadian oscillations also influence the microbial composition and functional profile during the 24-h period. Chronodisruption, including CLOCK mutant models, leads to dysbiosis and promotes obesity, while fasting, intermittent fasting or time-restricted feeding help re-set the balance. Multiple mechanisms are involved between these complex interactions [356,357,358,359]. Firstly, the species distribution of the microbes varies significantly between obese and lean individuals. The two dominant divisions of bacteria in human gut are *Firmicutes* and *Bacteroidetes*, and an increased *Firmicutes:Bacteroidetes* ratio has been associated with obesity [360]. High-fat diets and feeding during the biological night led to an increase in *Firmicutes*, while caloric restrictions, intermittent fasting, time-restricted feeding, plant-based, high-fibre diets, green tea, cranberry extracts, quercetin, resveratrol and persimmon tannins normalise this ratio and contribute to weight loss (Figure 4) [307,359,361,362,363,364,365,366,367]. 

Secondly, the gut flora is not a passive and silent community with actions limited to the gut; they are metabolically active and release metabolites into the gut, which are absorbed, enter host circulation and contribute to host physiology, including nutrient sensing. The dietary components and dietary patterns both define the microbial diversity, species distribution and microbial metabolites and have been linked to BMI, glucose homeostasis, metabolic disorders, several inflammatory and cardiovascular diseases, intestinal health, bio-activation of nutrients and vitamins, sleep disorders, neurodegenerative diseases and malignancy [368,369]. Plant-based diets, rich in fruit, vegetables, whole grains and polyphenols, promote microbial diversity, while Western diets, rich in fats and low in fibre, reduce gut microbial diversity [367]. One class of metabolites, called short-chain fatty acids (SCFAs), play multiple roles, including the regulation of energy derived from food, reducing inflammation, preventing pathogen invasion and maintaining barrier integrity. These also act as ligands to G-protein coupled receptors (GPCRs), which upon activation can lead to a variety of metabolic effects, including the stimulation of the secretion of insulin, glucagon-like peptide 1 (GLP-1) and peptide YY (PYY), which in turn, act to increase satiety and increase transit time [368,369,370,371]. Butyrate, a SCFA, also leads to AMPK activation and accompanying beneficial effects discussed above. Butyrate, a type of SCFA in the cecum of mice fed with regular chow, showed diurnal patterns with the ability to enhance the circadian gene amplitude in the liver, which was absent in high-fat-diet fed mice [372]. TRF and intermittent energy restriction caused the enrichment of species that upregulated the generation of SCFA [373]. The gut microbiota also contributes to the adipocyte phenotype and the expression of UCP1. Mice fed with a high-fat diet but over a limited feeding period showed an improved rhythmic expression of UCP1 and improved night-time energy consumption and oscillations in PPARα expression [232]. There is a reciprocal relationship between dietary polyphenols and the gut microbiota. Gut microbiota, as demonstrated by germ-free models, influence the metabolism and bioavailability of these compounds, while the health-promoting properties of polyphenols are partly mediated by their ability to act as a prebiotic to promote growth, reshape the microbial composition and reverse the disturbances in microflora caused by chronodisruption [137,163]. 

## 7. Conclusions 

Obesity is a critical global public health threat that without urgent and multifaceted approaches will continue its inexorable rise. Diet and energy balance remain key targets for any weight-regulatory intervention, but their application to manage long-term body weight have yielded limited success. The key to a successful dietary intervention requires sustained adherence with pragmatic implementation, such that there is minimal interference to day-to-day activities. Optimising circadian mechanisms via nutritional means is a valid and innovative approach to address energy balance. Additionally, educating and empowering the public to make informed lifestyle choices is key for sustainable behavioural changes. These modifications based on the appreciation of interactions between ‘what we eat’ and ‘when we eat’ can potentially prevent, delay the onset of and manage obesity. However, larger and longer human studies with clearly defined bioactive dietary components, timing of consumption, eating windows and energy content in genetically diverse populations are required before translating chrononutritional strategies into effective interventions for the general population.

## Figures and Tables

**Figure 1 nutrients-14-05080-f001:**
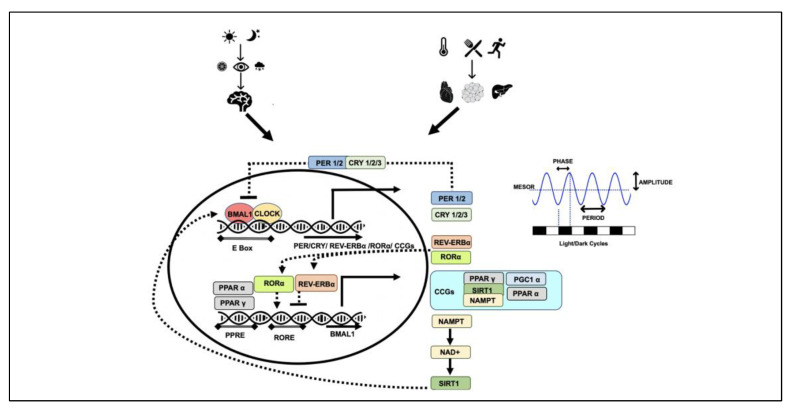
Circadian clock mechanisms: different zeitgebers lead to intricate transcription-translation feedback loop. circadian locomotor output cycles kaput (CLOCK) and brain and muscle Arnt-like protein-1 (BMAL1) heterodimerise and regulate the transcription of multiple clock-dependent genes (solid arrows). PER, CRY, REV-ERBα, RORα, PPARα, PPARγ, SIRT1 all lead to regulation of CLOCK and BMAL1 and contribute to their own regulation (dotted arrows). Amplitude, period, phase and MESOR of the oscillations produced determine the rhythmicity and robustness of the circadian clock. PER (Period), CRY (Cryptochrome), RORα (receptor-related orphan receptor α), REV-ERBα (NR1D1 gene producing a protein called REV-ERBα), PPARα (peroxisome proliferator-activated receptor α), PPARγ (peroxisome proliferator-activated receptor γ), NAMPT (nicotinamide phosphoribosyltransferase), NAD^+^ (nicotinamide adenine dinucleotide), AMPK (AMP-activated protein kinase), CCGs (clock-controlled genes), SIRT1 (sirtuin 1), PGC1α (PPARγ coactivator 1α), MESOR (midline estimating statistic of rhythm).

**Figure 2 nutrients-14-05080-f002:**
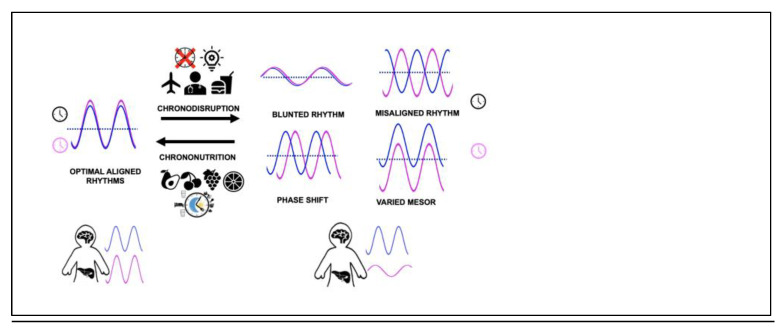
Master (blue) and peripheral clocks (pink) alignment is key to optimal metabolic health. Chronodisruption by various means leads to misalignment of circadian rhythms and has health consequences. Chrononutritional approaches have the ability to reverse deleterious chronodisruptive rhythms. The dotted line represents MESOR (midline estimating statistic of rhythm).

**Figure 3 nutrients-14-05080-f003:**
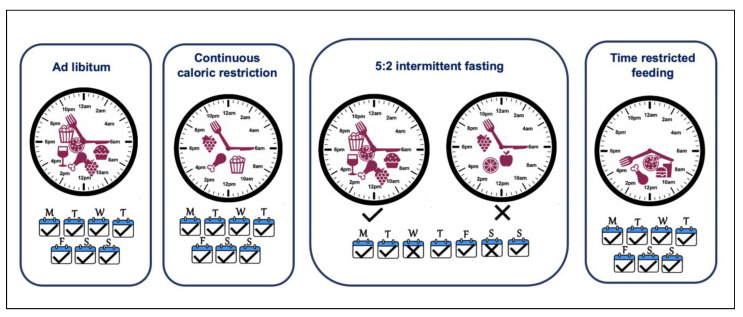
Dietary weight-loss approaches. Continuous caloric restriction includes reducing caloric intake on each day of the week but does not restrict time of the day. Intermittent fasting, e.g., 5:2 diet, introduces two caloric restriction days with ad libitum eating for the rest of the week. Time-restricted feeding limits the eating window and extended fast period regularly.

**Figure 4 nutrients-14-05080-f004:**
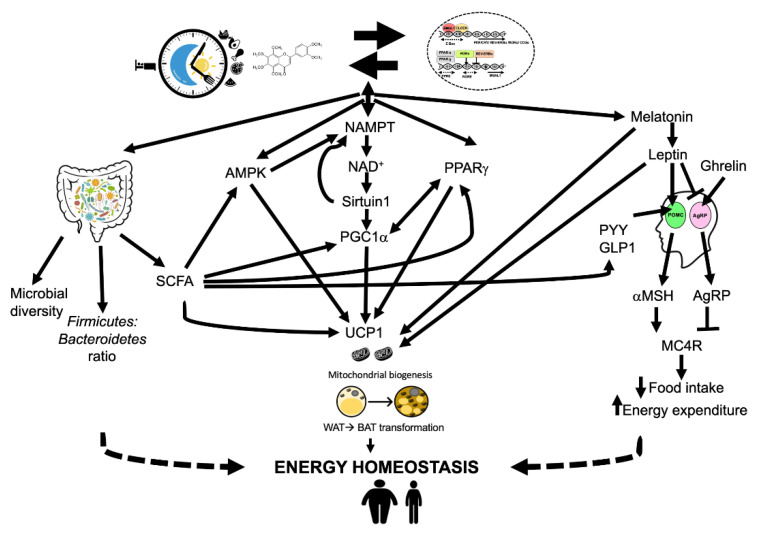
Mechanisms of chrononutrition: the bidirectional interactions between clock machinery, dietary polyphenols and meal times lead to rhythmic expression of various genes involved in sleep, energy balance, metabolism and regulation of gut microbiota. NAD, AMPK, SIRT1, SCFA, all derived from gut microbiota via PGC1a, lead to mitochondrial biogenesis and browning of white adipose tissue. MC4R pathway, via leptin, ghrelin, GLP1 and PYY, controls energy expenditure and satiety. Gut microbiota influenced by chrononutritional approaches also contributes to the overall energy homeostasis and BMI regulation. UCP1 (uncoupling protein 1) SCFA (short chain fatty acids), GLP1 (glucagon-like peptide-1), PYY (peptide YY), AgRP (agouti-related protein), POMC 9 (proopiomelanocortin), α-MSH (melanocyte-stimulating hormone), MC4R (melanocortin receptor 4), AMPK (AMP-activated protein kinase), NAD^+^ (nicotinamide adenine dinucleotide), PPARγ (peroxisome proliferator-activated receptor γ), NAMPT (nicotinamide phosphoribosyltransferase), NAD^+^ (nicotinamide adenine dinucleotide), AMPK (AMP-activated protein kinase), CCGs (clock-controlled genes), SIRT1 (sirtuin 1), PGC1α (PPARγ coactivator 1α).

**Table 1 nutrients-14-05080-t001:** A summary of genetic variants in clock components and their associations with chronotypes, eating behaviours and health parameters.

Genetic Variant/Haplotypes	Population	Observations	Ref
CLOCK rs1801260	421 Japanese subjects	C allele associated with an evening chronotype with significant delayed onset of sleep, shorter sleep times, greater daytime sleepiness	[52]
	500 overweight and obese subjects during 28-week weight-loss program	C allele carriers more emotional eaters and more resistant to weight loss	[53]
	284, 92 controls vs. 192 overweight and obese with or without binge eating disorder	C allele predisposes obese individuals to a higher BMI	[54]
	1272 overweight and obese participants attending 30-week weight-loss program	C allele carriers were emotional eaters and lost less weight and at a lower rate compared to non-emotional eaters with the same risk allele or non-risk allele	[55]
	370 children aged 6–13 years	No association between the genotype and sleep duration. However, there seems to be a trend between sleep duration and overweight	[56]
	85 overweight women	C allele carriers with significant circadian abnormalities: lower amplitude and greater fragmentation of the rhythm, and an evening chronotype	[57]
	1495 overweight/obese subjects participating in a Mediterranean-diet-based weight-reduction program for 28 weeks	C allele carriers are more resistant to weight loss, shorter sleep duration, higher plasma ghrelin concentrations, delayed breakfast time, evening preference and less compliance with a Mediterranean diet plan	[58]
	475 metabolic syndrome subjects participating in the CORDIOPREV clinical trial for 1 year	Gene diet interactions; C carriers showed non-significant improved insulin sensitivity while T carriers have significantly improved insulin sensitivity on low-fat diet	[59]
CLOCK haplotype rs1554483 and rs4864548	715 lean and 391 overweight or obese	GA haplotype associated with a 1.8-fold risk of overweight or obesity	[60]
CLOCK rs3749474	500 overweight/obese subjects	T allele carriers have significantly higher weight, BMI and waist circumference	[53]
CLOCK rs3749474	1100 individual participants in the Genetics of Lipid Lowering Drugs and Diet Network (GOLDN)	T allele carriers have significantly higher energy, total fat, protein and carbohydrate intakes	[61]
CLOCK rs3749474	898 subjects	T allele associated positively with higher BMI and evening carbohydrate intake	[62]
CLOCK rs6850524	260 cases with abdominal obesity and 260 controls Chinese population	CC genotype have a lower risk of overweight or obesity than those with GG genotype	[63]
CLOCK haplotype rs4864548-rs3736544-rs1801260	537 individuals from 89 families characterized for inflammatory, atherothrombotic and metabolic risk associated with insulin resistance.	CGC haplotype protective for the development of obesity and the CAT haplotype is associated with the presence of the metabolic syndrome	[64]
CLOCK rs4580704	7098 PREvención con DIeta MEditerránea (PREDIMED) trial over a median duration of 4.8 years	G allele with decreased incidence of type 2 diabetes and associated stroke. Mediterranean diet provides further protection in G allele carriers.	[65]
	897 patients from the CORDIOPREV clinical trial, low-fat diet and Mediterranean diet for 12 months	C allele carriers showed significant reduction in CRP levels and an improvement in HDL/ApoA1 ratio after a low-fat diet for 12 months	[66]
	1100 individual participants in the Genetics of Lipid Lowering Drugs and Diet Network (GOLDN)	G allele show lower blood pressure and higher erythrocyte membrane oleic acid (MUFA) and improved insulin sensitivity in high-MUFA intake	[67]
CLOCK haplotype rs3749474-rs4580704-rs1801260	1100 individual participants in the Genetics of Lipid Lowering Drugs and Diet Network (GOLDN)	The haplotype CGA was associated with lower BMI, weight, waist circumference, adiponectin concentration, blood pressure and with oleic acid (MUFA) RBC membrane composition.	[67]
CLOCK rs12649507, rs6858749	14,906 from the CHARGE (Cohorts for Heart and Aging Research in Genomic Epidemiology) Consortium	Longer habitual sleep duration is associated with lower BMI and a favourable dietary behaviour. rs12649507 G allele: Higher PUFA intake and more sleep.rs6858749 T allele: Lower protein intake with each additional hour of sleep	[68]
CLOCK rs10002541 and rrs6850524	260 cases with abdominal obesity and 260 controls, Chinese population	Significant associations between CG and TG haplotypes and abdominal obesity. rs10002541 C allele is protective for abdominal obesity.	[63]
CLOCK rs12649507 and rs11932595	1011 individuals from Tyrol and Estonia	The haplotype GGGG in Tyrolean and GGAA in Estonian population associated with longer sleep	[69]
CLOCK rs1801260 BMAL1 rs2278749	507 healthy young adults	CLOCK rs1801260 associated with seasonal affective disorder and it synergistically interacts with BMAL1 rs2278749. Seasonal variation in body weight and appetite also associated with BMAL1 rs2278749.	[70]
CLOCK rs1801260 + SIRT1 rs1467568	1465 Overweight/obese subjects over 30 weeks of Mediterranean-diet-based weight-loss program	SIRT1 (A allele) ad CLOCK (C allele) are more resistant to lose weight and have more of an evening preference	[71]
PER2 gene rs4663302 rs2304672	454 obese subjects, weight reduction program based on Mediterranean diet	PER2 gene rs4663302 rs2304672 associated with abdominal obesity. rs4663302 T allele and rs2304672 G allele carriers associated with not completing the weight-loss programme. rs2304672 G also linked to extreme snacking, experiencing stress with dieting, eating when bored, and skipping breakfast than noncarriers.	[72]
PER2, BMAL1, and NPAS2	189 patients with winter depression and 189 matched controls	PER2 rs10870, BMAL1 rs2290035 and NPAS2 rs11541353 significantly association with SAD	[73]
CRY1 rs10861688	260 cases with abdominal obesity and 260 controls, Chinese population	CRY1 rs10861688 T allele negatively associated with the risk of abdominal obesity.	[63]
REV-ERBα rs2314339	2212 subjects from two independent populations (1402 from Spanish Mediterranean and 810 North American)	Minor allele carriers (AA + AG) have lower probability of abdominal obesity than noncarriers. A allele carriers on low MUFA lead to high BMI while A carriers on high MUFA reduce BMI and BMI was low in A carriers in high-PUFA intake	[74]

CLOCK (circadian locomotor output cycles kaput) BMAL1 (brain and muscle Arnt-Like protein-1) PER (Period), CRY (Cryptochrome), REV-ERBα (NR1D1 gene producing a protein called REV-ERBα), MUFA (mono-unsaturated fatty acid), SAD (Seasonal affective disorder), NPAS2 (Neuronal PAS Domain Protein 2).

**Table 2 nutrients-14-05080-t002:** A summary of evidence establishing the role of chronodisruption by mistimed eating habits and associated BMI and metabolic health outcomes.

Late Night Eating, BMI and Metabolic Health		
180 bariatric surgery candidates, 93 non-surgical weight-loss intervention and 158 general community candidates	Night-eating syndrome associated with binge eating, higher BMI and male gender. Night-eating syndrome, consuming nocturnal snacks leads to more hunger and depressive symptoms.	[87]
26,902 men over 16 years follow-up	Late night eaters have a 55% higher risk of CHD, are more likely to have baseline hypertension and men who did not eat breakfast have a 27% higher risk of CHD than those who ate breakfast.	[90]
8153 adults over an average of 3.9 yrs.	Night-time eating associated with dyslipidaemia in both men and women but metabolic syndrome and an increase in the risk of obesity only in women	[96]
10 participants on glucose solution at 8 am and 8 pm and 9 participants taking a low-glycaemic-index meal at 8 am, 8 pm and midnight.	Even low-glycaemic-index meals late at night disturb glucose metabolism	[86]
19,687 Japanese women	Skipping breakfast, late dinner and bedtime snack associated with overweight and obesity in Japanese women	[97]
397, 8–12 yr old children	Late dinner eaters (after 21.07 h) were more likely to be overweight and obese, with higher waist circumference and inflammatory markers	[98]
49 participants	Inconsistent meal time, especially late, eaters have significantly higher BMI	[99]
100 subjects	Correlation between night-time eating and binge eating and BMI	[88]
**Later chronotype, BMI and metabolic health**		
2200 9–16 yr old in Australia	The later chronotype more likely to be overweight and obese	[100]
54 college freshmen	An evening chronotype associated with higher BMI as compared to morning or neutral chronotypes	[49]
511 UK 11–13 yr old children	An evening chronotype was associated with higher BMI, higher frequency of unhealthy food choices. Sleep duration is an independent risk factor for BMI	[101]
194 participants	Later chronotype and larger dinner are associated with poorer glycaemic control in patients with type 2 diabetes independently of sleep disturbances.	[45]
439,933 adults from a UK biobank	Evening preference associated with high risk of cardiovascular disease	[46]
800 undergraduate students	Evening chronotype associated with BMI that can be negated by a decrease in sugary beverage intake, increases in physical activity	[47]
2133 prediabetic patients	More evening preference is directly associated with higher BMI	[48]
872 adults	Later chronotypes with higher percentage of daily energy intake during the night are associated with overweight and obesity while earlier chronotypes consuming more energy in the morning are at a lower risk of weight gain.	[102]
1197 middle-aged men and women	An evening chronotype associated with obesity	[50]
**Circadian misalignment, BMI and metabolic health**		
14 healthy participants on 8-day protocol for short-term misalignment and cross over	Short-term circadian misalignment leads to increased systolic and diastolic blood pressure and serum inflammatory markers	[103]
14 adults in a 6-day simulated shift-work environment	Eating during the biological night, e.g., for shift workers, decreases total daily energy expenditure and increases the risk of weight gain and obesity	[89]
10 adults underwent a 10-day protocol with eating and sleeping in all phases	Forced desynchrony protocols cause circadian misalignment and disturb postprandial glucose response typical of prediabetes	[104]
2494 participants (1259 day and 1235 shift workers)	Shift work is associated with higher risk of being overweight/obese	[105]
26,382 participants (9088 shift workers)	Long-term shift work is associated with metabolic syndrome	[106]
9912 male employees (8892 daytime workers and 920 rotating three-shift workers	High risk of obesity among male shift workers	[107]
905 shift workers	Strong association between sleep deprivation and obesity in shift workers	[108]
200 shift workers	Night work is a risk factor for abdominal obesity, social jetlag is higher in night shift workers and it was associated with the presence of obesity.	[109]
3188 shift workers and 6395 non-shift workers	Shift work associated with obesity, lower physical activity, poor dietary choices	[110]

**Table 3 nutrients-14-05080-t003:** A summary of polyphenols with their interactions with circadian system and role in health.

Polyphenol	Interactions with Circadian Clocks	Reference
Resveratrol	Improvement in rhythmic expression of core clock and various clock-controlled genes including NAMPT, SIRT1, PPARα	[142,143,144,145,146,147,148]
	Improvement of insulin, glucose, lipid metabolism impairments caused by chronodisruption	
	Prevention of neuron damage and memory impairment caused by circadian disruption	
	Enhanced expression of PER1, PER2 and BMAL1 in rat fibroblasts	
	Downregulation of high-fat diet induced REV-ERBα in adipose tissue	
	Reversal of free-fatty-acid-induced loss of oscillation amplitude in core clock genes in HEPG2 cells	
	Amelioration of acrylamide suppressed amplitude and phase of oscillations in core clock genes and increased expression of SIRT1 and PGC1α	
	Increase in expression of BMAL1, PER1, SIRT6, SIRT1 and REV-ERBα mRNA in fibroblasts	
Proanthocyanidins	Modulation BMAL acetylation, increase in PER2 expression and inhibition of REV-ERBα and RORα in rat models of diet-induced obesity	[149]
	Modulation of NAMPT expression and NAD^+^ levels in rat liver	
Epigallocatechin-3-gallate	Altered circadian expression patterns of CLOCK, BMAL1 and key appetite-regulating genes in mice	[150,151,152]
	Ameliorated diet-induced metabolic misalignment by regulating the rhythmic expression of the circadian clock genes in the liver and adipose tissue in mice	
	Repressed CLOCK expression in lung cancer cell lines and reduced the self-renewal capacity of the cells	
Nobiletin	Affects amplitude, period and phase of mutant mice cells with weaker rhythmic amplitude.	[153,154,155,156,157,158,159,160,161,162]
	Activation of RORs and protection against metabolic syndrome in a clock-dependent manner.	
	Enhances BMAL1, reverses the loss of oscillation amplitude observed in metabolic disease state	
	Induction of AMPK-SIRT1 signalling and lipogenesis	
	Ability to induce circadian rhythmicity and inhibit oncogenicity in MDA-MB-231 cells	
	Modulates expression of core clock and clock-controlled genes in the cortex	
	Restores endogenous rhythm of clock genes in steatosis liver	
	Modulates clock and Alzheimer’s-disease-related genes in cortex of AD model mice	
	Cardioprotective role in ischemia reperfusion injury by upregulating midazolam-inhibited PER2	
	Improves metabolic fitness in naturally aged mice and promotes healthy aging in high-fat diet by activating genes to promote mitochondrial function	
	Improves insulin secretion by enhancing the amplitude of circadian gene expression in T2D islets	
Oolong tea polyphenols	Improves *Firmicutes:Bacteroidetes* ratio in the intestinal flora in mice	[163,164]
	Promotes the growth of strains of gut microbiota and positively regulates the production of SCFA	
	Restoration of CLOCK, BMAL1, PER and CRY expression disturbed by constant day feeding in mice	
Chichoric acid	Regulation of rhythmic expression of clock genes	[165]
Quercetin	Upregulation of BMAL1, SIRT1, SIRT6, REV-ERBα and reduction in PER1 expression in fibroblasts	[148,166]
	Suppresses breast cancer metastasis to lymph nodes promoted by circadian disruption	

CLOCK (circadian locomotor output cycles kaput), BMAL1 (brain and muscle Arnt-like protein-1), PER (Period), CRY (Cryptochrome), RORα (receptor-related orphan receptor α), REV-ERBα (NR1D1 gene, producing a protein called REV-ERBα), NAMPT (nicotinamide phosphoribosyltransferase), SCFA (short chain fatty acids), T2D (type 2 diabetes), SIRT6 (Sirtuin 6), SIRT1 (sirtuin 1), AMPK (AMP-activated protein kinase).

**Table 4 nutrients-14-05080-t004:** A summary of human studies investigating the impact of time-restricted eating in humans and metabolic health outcomes.

Time-Restricted Feeding (TRF)					
Participants	Eating Restrictions	Study Type/Duration	Health Outcomes	Energy Intake	Reference
*n* = 49 obese subjects BMI 30–50 kg/m^2^	Eating window of 4 h (3 pm–7 pm) vs. 6 h (1 pm–7 pm) vs. controls (7 am–7 pm)	Randomized parallel-arm trial over 8 weeks	Both TRF regimens showed reduction in body weight, insulin resistance, oxidative stress levels. Four-hour TRE did not result in greater weight loss compared to six-hour TRE.	Reduction in energy intake by 550 kcal/day in both cases without calorie counting	[176]
*n* = 11 Obese sedentary males BMI: 30.2–34.2 kg/m^2^	Eating window 10 am–5 pm vs. 7 am–9 pm	Randomized crossover trial; 3 weeks each intervention of 5 days with 10 days washout period	Improved glycaemic control and decrease in evening hunger	Isocaloric intake	[177]
*n* = 19 with T2D BMI: 29–39 kg/m^2^	4-week TRE 10 am–7 pm	non-randomised 2-week baseline, 4-week intervention	Compliance 72 ± 24%, no improvement in glycaemic control or body mass	Isocaloric intake	[178]
*n* = 23 obese subjects BMI 30 and 45 kg/m^2^	Eating over 8-h window (10 am–6 pm) vs. ad libitum eating	2-week baseline intake, 12-week intervention	Time-restricted eating showed reduction in body weight and systolic blood pressure	Decreases caloric intake by ~300 kcal/d	[179]
*n* = 34 resistance-trained weight 84.6 ± 6.2 kg	TRF (1 pm–8 pm) vs. control (8 am–8 pm)	Randomized parallel-arm trial over 8 weeks	TRF only showed a reduction in fat mass but no other metabolic parameters were altered. Fat free mass and muscle mass area in arm and thigh remain unchanged	Isocaloric intake	[180]
*n* = 9 overweight sedentary older adults BMI 25–40 kg/m^2^	16 h fast (14–18 h range)	Baseline assessment followed by 4-week intervention	TRE resulted in short-term weight loss and improved waist circumference, cognitive and physical function and health-related quality of life	No data available	[181]
*n* = 8, prediabetic BMI 32.2 ± 4.4 kg/m^2^	eTRF; 6-h eating period and dinner before 3 pm for 5 weeks, vs. 12-h eating period	Randomized crossover trial for 17 weeks, each intervention 5 weeks	eTRF reduced insulin levels and improved insulin sensitivity, lowered blood pressure; reduction in oxidative stress and appetite in the evening.	Isocaloric intake	[182]
*n* = 19 with metabolic syndrome	Eating over self-selected 10-h window	2-week baseline intake, 12-week intervention	TRE improves cardiometabolic health (reduction in weight, BMI, waist circumference, percentage body fat, systolic and diastolic blood pressure, improved lipid parameters, glucose and insulin homeostasis	Decreases caloric intake	[183]
*n* = 8 overweight BMI > 25 kg/m^2^	Eating over self-selected 10-h window	3-week baseline intake, 16-week	Reduction in body weight. Significant improvement in sleep, hunger at bedtime, energy levels	Reduced estimated energy intake by 20–26%	[170]
*n* = 11, BMI 25.0 and 35.0 kg/m^2^	eTRF (8 am to 2 pm) vs. control (8 am to 8 pm)	Randomized crossover 4-day intervention, 3.5–5 weeks’ washout period between interventions	eTRF improves 24-hour glucose levels, alters lipid metabolism and expression of SIRT1 and LC3A (autophagy gene), BDNF (a neurotrophic factor promoting neuronal growth) and mTOR	Isocaloric intake	[184]
*n* = 11 overweight BMI 25–35 kg/m^2^	eTRF (8 am–2 pm) vs. control (8 am–8 pm)	Randomized crossover 4-day intervention, 3.5–5 weeks’ washout period between interventions	Meal-timing interventions facilitate weight loss primarily by decreasing appetite rather than by increasing energy expenditure. eTRF may also increase fat loss by increasing fat oxidation.	Isocaloric	[185]
*n* = 21 healthy adults BMI 29.6 ± 2.6 kg/m^2^	TRE (12 pm to 8 pm) vs. control eating habits with concomitant aerobic exercise for 8 weeks	Randomized, controlled trial	TRF individuals lost significantly more body mass (3.3% vs. 0.2%) and fat mass (9% vs. 3.3%). Lean mass increased but no significant difference between the groups.	Reduction in caloric intake in TRE (~300 kCal/day)	[186]
*n* = 27 BMI 21.9–26.9 kg/m^2^	TRE included an elimination of caloric intake between 7 pm and 6 am vs. controls	Crossover 2-week intervention with one-week washout period	TRE led to a loss in small amount of body weight	Reduction in energy (~240 Kcal) and fat intake un TRE group	[187]
*n* = 18 Body weight 79.0 ± 13.5 kg in control group and 87.4 ± 19.2 in TRE group	TRE (eating over any 4-h window between 4 pm and 12 pm on the four days a week when they exercised but ad libitum on days without exercise) vs. control without any restrictions	Randomized controlled trial 8 weeks	No significant loss of body weight, no adverse effect on lean mass retention or muscular improvements.	TRF reduced caloric intake by ~667 kCal a day	[188]
*n* = 13 BMI 20–39 kg/m^2^	TRF with delayed breakfast and advanced dinner by 1.5 h vs. controls with habitual eating patterns	2-week baseline, 10 weeks’ intervention	No significant reduction in weight, but reduction in adiposity, fasting glucose observed	Reduction in energy intake in TRE group	[189]
*n* = 40 resistance-trained females Body weight 57.1 to 73.4 Kg	Control diet vs. TRF (12 pm–8 pm) vs. TRF+ a leucine metabolite β-hydroxy β-methyl butyrate (HMB) supplementation	randomized, placebo-controlled for 8 weeks	TRF did not produce changes in physiological variables including resting metabolic rate, substrate utilization, blood lipids, glucose and insulin, blood pressure, arterial stiffness, or cortisol responses. No significant difference in physical performance.	No significant variation between the groups	[190]
*n* = 40 with abdominal obesity BMI 25.1–37.6 kg/m^2^	TRF eating window 8–9 h	3-month single arm trial	Moderate weight loss, improved waist circumference, HbA1C	No data available	[191]
*n* = 22 men BMI: 28.5 ± 8.3 kg/m^2^	Isocaloric TRF (8-h eating window, caloric intake within 300 Kcal of habitual intake) vs. ad libitum TRF (8-h eating window but no restriction on calories)	28 days randomised control trial	Decrease in body mass, decrease in fat body mass, decrease in BP and increase in HDLC in both groups.	No significant difference in caloric intake	[192]
*n* = 20 obese BMI 34.1± 7.5 kg/m^2^	TRE (self-selected 8-h eating window) vs. control on ad libitum	12 weeks	Decrease in eating frequency, weight, lean mass, visceral fat	No data reported	[193]
*n* = 116 overweight and obese BMI 27.4–35.4 kg/m^2^	TRE (12–8 pm) eating vs. ad libitum	12 weeks’ randomised control trial	Loss of body weight in TRE group	No significant difference in caloric intake	[194]
*n* = 271 NAFLD BMI > 24 kg/m^2^	Control vs. ADF (25% energy intake on fast days) vs. TRF (self-directed 8-hour window)	12 weeks’ randomised control trial	Significant weight loss and fat mass loss, reduction in cholesterol and triglycerides both in ADF and TRF with ADF achieving better outcomes	No significant difference in caloric intake	[195]
*n* = 15 PCOS women BMI ≥ 24 kg/m^2^	TRE (8 am–4 pm)	Non-randomized 1 week baseline, 5 weeks’ intervention	Reduction in body weight, BMI, body fat mass, body fat percentage, improved insulin sensitivity	Isocaloric	[196]
*n* = 22 BMI = 24.7 ± 0.6 kg/m^2^	TRE (eating within 8-h window but first meal between 10–11 am) vs. controls with normal feeding patterns	Randomized controlled trial, 1 week baseline, 6-week intervention	No weight loss or improvement in cardiovascular function with modest improvement in functional endurance and glucose tolerance, 84–95% adherence	Isocaloric	[197]
*n* = 60 BMI ≥ 30 kg/m^2^	14:10 TRE (14-h metabolic fast with snack with 200 kcal mixed nuts 12 h after the fast started) vs. 12:12 TRE (12-h fast without any snack)	Randomized controlled trial 8 weeks intervention	Weight loss observed in both cases but more in 14 h metabolic fast group, improved fasting blood glucose. Fasting snack decreased hunger	500–1000 kcal deficit each day	[198]
*n* = 45 with at least one metabolic syndrome component and usual eating window of 14 h. BMI ≥ 20 kg/m^2^	TRE (self-selected window of 12 h) vs. no restriction	Randomised control trial 4 weeks’ baseline, 6-month intervention	No significant difference in weight loss	No difference reported	[199]
*n* = 80 males	TRF (8-h eating window, 7.30 pm–3.30 am) vs. normal diet daily fasting for 16 h for 25 days	25 Days	TRF improved lipid parameters, reduced inflammatory markers, enhanced gut microbial richness with enrichment of *Prevotellaceae* and *Bacteroideaceae*; activated SIRT1	No data available	[200]
**Time-restricted feeding (TRF) vs. Continuous caloric restriction (CR)**					
*n* = 16 BMI 24.0 ± 0.6 kg/m^2^	eTRF (8 am till 4 pm) vs. control on caloric restriction	Non-randomised 1-week baseline, 2 weeks’ intervention	eTRF improved insulin sensitivity, glucose uptake, reduction in energy intake and weight loss	Isocaloric	[201]
*n* = 37 overweight BMI 26.4–28.55 kg/m^2^	TRE (8 am–4 pm) vs. BMI matched participants on hypocaloric diet based on orthodox fasting		Both groups showed reduction in BMI and fasting group also showed a reduction in total and LDL cholesterol	Isocaloric	[202]
**Early time-restricted feeding (eTRF) vs. late time-restricted feeding (lTRF)**					
*n* = 15 men at risk of T2D BMI 33.9 ± 0.8 kg/m^2^	eTRF (8 am–5 pm) vs. late TRF (12 pm–9 pm) over 2-time 7-day TRF with 2 weeks’ washout period	Randomised crossover trial, 1-week baseline, 1-week intervention, 2 weeks’ washout period	Both TRF regimens showed reductions in body weight, glycaemic responses to a test meal, triglycerides.	No data available	[203]
*n* = 82 BMI 18.6–25.8 kg/m^2^	Early TRF (6 am–3 pm) vs. mid-day TRF (11 an–8 pm) vs. controls	Randomised control trial, 5 weeks’ intervention	Early TRF was more effective in improving insulin sensitivity, fasting glucose, reduction in body mass and adiposity, reduction in inflammation and increased gut microbial diversity	Reduction in caloric intake in both TRF groups vs. control	[204]
*n* = 8 prediabetic BMI 32.2 ± 4.4 kg/m^2^	eTRF (6-h eating window and dinner before 3 pm) vs. 12-h eating period	Randomised crossover trial, 5 weeks’ intervention with a 7-week washout period	eTRF reduced insulin levels and improved insulin sensitivity, lowered blood pressure, reduction in oxidative stress and appetite in the evening.	Isocaloric	[181]
**Breakfast vs. dinner calories**					
*n* = 93 Overweight and obese women BMI 32.4 ± 1.8 kg/m^2^	1440 KCal consumed over breakfast/lunch/dinner 700, 500, 200 kcal vs. 200,500, 700 kcal	Randomized parallel-arm study for 12 weeks.	High caloric breakfast group showed greater weight loss and waist circumference, fasting glucose, insulin, triglycerides, HOMA-IR. Ghrelin, hunger vs satiety improved.		[205]
*n* = 1245 non-obese, non-diabetic middle-aged adults	Daily caloric intake at dinner (<33% vs. 33–48 vs. ≥48% of daily kcal)	6 years	Consuming more calories at dinner is associated with an increased risk of obesity, metabolic syndrome and NAFLD		[206]

TRF (time-restricted feeding), TRE (time-restricted eating), NAFLD (non-alcoholic fatty liver disease), HOMA-IR (homeostatic model assessment for insulin resistance), HbA1C (haemoglobin A1C), BDNF (brain-derived neurotrophic factor).

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
