# Peer review of "Chrononutrition—When We Eat Is of the Essence in Tackling Obesity"

_nutrients, 2022, doi:10.3390/nu14235080_

Round 1

Reviewer 1 Report

This manuscript is a thorough review of current scientific evidence that can help understand the impact of chrononutrition (when we eat) in energy homeostasis and obesity.  The review is of considerable merit, representing an important contribution to the literature. The review is extensive and complete and authors have done an excellent job organizing the manuscript for an easy comprehension. Nevertheless, there are some minor comments that the authors need to address several issues prior to publication:

1.       The authors should mention in the manuscript the evidence, in animal models, of the effects of chrono-nutritional interventions on clock gene expression (epigenetics)

2.       Please include reference of the study mentioned 312 page 11 (regarding the use of tryptophan enriched milk formula in infants at different times of the day)

Additional minor issues include the following.

1.       It would be recommended that the format of table 1 respects the genetic variants in the first column, in order that they do not have a number posted in the next row

2.       First row of table 2, the letter should not be bold. First two rows of the first column of table 3 should not be bold either

3.       References in table 2 and 4 should be moved to the last column to continue the same design used in the other tables

4.       Page 22 starts again as page 1

5.       Correct the parts of the text that are unnecessarily underlined (row 282 page 11; row 494 to 499; row 550; row 565-566; row)

Author Response

  1. The authors should mention in the manuscript the evidence, in animal models, of the effects of chrono-nutritional interventions on clock gene expression (epigenetics) This has been already been addressed in the mechanisms section. 
  2. Please include reference of the study mentioned 312 page 11 (regarding the use of tryptophan enriched milk formula in infants at different times of the day) This has been added
  3. It would be recommended that the format of table 1 respects the genetic variants in the first column, in order that they do not have a number posted in the next row I don't understand what the reviewer means by the number posted in the next row 
  4. First row of table 2, the letter should not be bold. First two rows of the first column of table 3 should not be bold either Amended 
  5. References in table 2 and 4 should be moved to the last column to continue the same design used in the other tables Amended 
  6. Page 22 starts again as page 1 For some reason, I can't amend it but I suppose the publishers will take care of it 
  7. Correct the parts of the text that are unnecessarily underlined (row 282 page 11; row 494 to 499; row 550; row 565-566; row) Amended

Reviewer 2 Report

This paper is of good quality. It give insights on the chrono-nutrition and the impact of the ingestion's time on human health. The manuscript is good but required some modifications.  

Please see my comments on the manuscript.

Author Response

  1. Line 53: add a reference for the statement of chrononutrition being a promising target for weight management therapies. This is a novel target and a substantial body of research indicates in its role in health promotion. This article is about exploring this further, hence no reference.
  2. line 80: MNTR1 and MNTR2: full names: added
  3. Table 1: is there any impact of gender? No, hence it has not been mentioned
  4. Table 2: Table is misleading? And Late night eating: what is the dependent variable. I don’t understand what this means; I have addressed the dependent variable and hopefully it is clearer now. This table supports the fact that eating patterns not aligned to chronobiogy are detrimental in terms of BMI and metabolic health.
  5. Line 284: elaborate the impact of the nutrients mentioned on circadian biology: more detail and two references have been added
  6. Line 295: what is the food source of melatonin: added
  7. Line 317: dietary sources of melatomin: added
  8. Line 326: two paragraphs same title: I don’t get what you mean by this; one is about ‘what’ and the other is about ‘when’.
  9. Line 326, rephrase the sentence: I feel the sentence is okay as it is.
  10. Diagrams need to be of improved quality: an effort has been made to improve the clarity.